# Culture on Selective Media and Amplicon-Based Sequencing of 16S rRNA from Spontaneous Brain Abscess—the View from the Diagnostic Laboratory

Camilla Andersen,[a] Bo Bergholt,[b] Winnie Ridderberg,[a] [ID] Niels Nørskov-Lauritsen[a]

[a]Department of Clinical Microbiology, Aarhus University Hospital, Aarhus, Denmark
[b]Department of Neurosurgery, Aarhus University Hospital, Aarhus, Denmark

**ABSTRACT** Forty-one stored samples from cases of spontaneous brain abscess were investigated to gain insight into the natural history, causative agents, and relevant laboratory diagnostics of a rare infection. Samples from a larger collection were selected based on retrospective analysis of patient records. All samples were subjected to amplicon sequencing of 16S rRNA gene fragments. Supplementary culture on selected media was performed as suggested by bioinformatics analysis. For three cases, no microorganism was disclosed, while *Toxoplasma gondii*, *Aspergillus fumigatus*, and various bacteria were the cause of 1, 2, and 35 cases, respectively. Bacterial infections were monomicrobial in 20 cases and polymicrobial in 15; the microorganisms of the latter cases were restricted to residents of cavum oris. Amplicon sequencing did not further enhance the importance of the *Streptococcus anginosus* group, which was involved in 17 cases, and the single primer set used may be suboptimal for amplification of *Actinomyces* and *Nocardia*. But, amplicon-based sequencing unquestionably expanded the number of polybacterial infections, with focus on the *Fusobacterium nucleatum* group, *Parvimonas*, and *Porphyromonas*. Culture on selective media confirmed the presence of *F. nucleatum* group bacteria, which attained a prominence in spontaneous brain abscess similar to the *S. anginosus* group. Metagenomics is a powerful tool to disclose the spectrum of agents in polymicrobial infections, but a reliable cutoff value for substantial detection is complex. Commercial media for isolation of *F. nucleatum* group bacteria from mixed infections are available, and these pathogens should be carefully characterized. Isolation of *Parvimonas* and *Porphyromonas* in polymicrobial infections has not been resolved.

**IMPORTANCE** Polymicrobial brain abscess is a challenge to the clinical microbiology laboratory due to the aggregative nature of the dental and oral microbiota. Because polymicrobial infections may escape detection by conventional culture methods, directed therapy toward a single detected bacterium is problematic. Amplicon-based sequencing provides important clues to these infections, but only cultured microorganisms can be fully characterized, subjected to antimicrobial susceptibility testing, and formally named. By use of specific selective culture plates, we successfully isolated bacteria of the *Fusobacterium nucleatum* group, and these bacteria rose to the same prominence as the widely recognized pathogen, the *Streptococcus anginosus* group. Named and unnamed members of the *Fusobacterium nucleatum* group must be further investigated to gain insight into a rare but grave disease.

**KEYWORDS** *Fusobacterium*, odontogenic infection, oropharyngeal microbiota, *Parvimonas*, polymicrobial

Address correspondence to Niels Nørskov-Lauritsen, niels.norskov-lauritsen@rsyd.dk.

The authors declare no conflict of interest.

**B**rain abscess is a rare but grave infection with a 1-year mortality of approximately 20% (1, 2). Examination of medical records of Danish patients with first-time International Classification of Diseases 10th Revision code for brain abscess from 2007

to 2016 revealed an incidence rate of 0.78 per 100,000 person-years (3). Brain abscess formation may occur after accidental or neurosurgical head trauma, by contiguous spread from parameningeal foci, or by hematogenous spread of microorganisms from distant or cryptogenic foci (4, 5). Contiguous infections have been defined as infections secondary to otitis media, mastoiditis, sinusitis, or dental infections (1). Therapy encompasses surgical drainage with stereotactic aspiration and prolonged treatment with antimicrobial agents.

Bacteria are cultured from 70 to 90% of specimens (1, 6, 7). *Staphylococcus aureus* is usually associated with previous neurosurgery or trauma, while *Streptococcus* species predominate in spontaneous abscesses. Less frequent bacterial pathogens include *Nocardia* and *Actinomyces*. Fusiform bacteria in brain abscesses were recorded by McFarlan as early as 1943 (8), but the common notion of "microscopy only" in that paper testifies to the challenge of culture of polymicrobial infections. In a Danish study from 1994 to 2009, pus was aspirated from 89 of 102 cases (1). Gram-stain and microscopy indicated polymicrobial infection in 23 specimens, but mixed infections were only corroborated by culture in 11 cases. Reviews of causative bacterial agents of brain abscess are available (5, 9), but developments of diagnostic and therapeutic procedures, as well as nomenclatural changes, impede historical comparisons.

Characterization of the microbiota has been attempted by PCR amplification of 16S rRNA genes (10–13) (Table 1). The Marseilles group investigated 39 samples, which were positive for 16S by conventional small-fragment PCR, by amplification of almost full-length 16S rRNA and subsequent cloning and sequencing of 100 amplicons from each patient. Probably due to previous antimicrobial therapy, a large number of samples were culture-negative, and monomicrobial culture with *Cutibacterium acnes* and *Streptococcus pneumoniae* was frequent (and verified by sequencing). Nineteen specimens were polymicrobial, and two distinct bacterial populations from dental and sinusal origin were discriminated (11). A nationwide study from Norway applied group-specific PCRs combined with Sanger sequencing and software for analysis of mixed sequencing chromatograms, enabling detection of up to 9 bacterial species per sample (12). The analysis allowed for triple the number of bacterial identifications. *Aggregatibacter aphrophilus*, *Fusobacterium nucleatum*, and *Streptococcus intermedius* or combinations of them were found in all polymicrobial abscesses. Conventional culture in that study was extensive but failed to a large extent to corroborate DNA traces of *Eubacterium*, *Campylobacter*, and *Parvimonas*. A recent study from Germany investigated stored DNA samples by use of the 16S metagenomic protocol of Illumina that targets the V3 + V4 region (13). Only successful amplifications (as evaluated by gel electrophoresis) were subjected to next-generation sequencing. A total of 86 bacterial taxa were identified, with *S. intermedius* and *F. nucleatum* being the most prevalent species. The threshold employed to discriminate true-positive reads allowed for identification of up to 16 bacterial taxa per sample, and half of the samples were polymicrobial (Table 1).

We investigated stored intracerebral specimens by amplicon-based sequencing of 16S rRNA, and performed supplemental tests and directed culture as suggested by bioinformatics analysis. Inclusion criteria were cerebral samples received for culture plus retrospective analyses of patient records to identify cases of spontaneous brain abscess.

## RESULTS

**Patients and conventional culture.** Medical records from 121 patients who had brain specimens submitted for microbiology culture from September 2011 through 2017 were reviewed by a neurosurgeon and an infectious disease specialist. Excluding empyema, tumor, hemorrhage, and patients with recent brain surgery, 41 cases (37% female) were categorized as spontaneous brain abscess, corresponding to an incidence rate of 0.51 per 100,000 person-years. The mean and median age were 54 and 59 years, respectively (range, 0 to 82 years), and mortality after 30 days and after 1 year was 2.4 and 15%, respectively. Patient characteristics, Gram microscopy, initial culture, and microbial diversity by amplicon sequencing of 16S rRNA genes are outlined in Table 2. Four specimens were culture negative, while filamentous fungi (*Aspergillus fumigatus*)

**TABLE 1** Case series of pyogenic brain abscess investigated by sequencing of 16S rRNA gene fragments

| Study (reference) | No. of patients (period) | Disease categorization (no.) | Inclusion criteria | Material | Primer sets | E. coli range (excluding primers) | Sequencing | Avg depth[a] | No. polymicrobial infections (%) | No. bacterial taxa identified |
|---|---|---|---|---|---|---|---|---|---|---|
| France (11) | 39 (2006–2010) | Intracerebral lesions (33); postoperative infections (6) | Samples positive by conventional 16S PCR | Stored DNA | One pair | nt 28–1491 (1,464 nt) | Cloning (Sanger) | 100 | 19 (49) | 76 |
| Norway (12) | 31 (2011–2013) | Spontaneous brain abscess | Samples positive by conventional 16S PCR and/or culture | Stored DNA | Three sets | nt 62–320 (V1 + V2, 259 nt) | Massive parallel (Sanger/ion torrent) | 245,600 | 22 (71) | 47 |
| Germany (13) | 35 (2010–2016) | Brain abscess (31); epidural infections (4); few clinical data | Samples positive by conventional 16S PCR and/or culture | Stored DNA | One pair | nt 358–784 (V3 + V4; 427 nt) | NGS (Illumina 16S metagenomic sequencing) | 20,000 | 18 (51) | 86 |
| Present study | 41 (2011–2017) | Spontaneous brain abscess | Cases retrospectively classified as spontaneous brain abscess | Stored abscess material | One pair | nt 358–784 (V3 + V4, 427 nt) | NGS (Illumina 16S metagenomic sequencing) | 154,247 | 15 (43) | 18 to 60 genera, depending on cutoff |

[a]Investigated clones or aligned reads from individual specimens.

**TABLE 2** Patient characteristics, microbiota, and outcome of 41 cases of spontaneous brain abscess

| Patient no. | Sex/age | Risk factor[d] | Pre-OP AB | Gram stain[c] | Culture | 16S sequencing (% of aligned reads) (genus or group level) | Aligned reads[d] | Outcome at 30 day/1 yr[e] |
|---|---|---|---|---|---|---|---|---|
| 1 | F/5 | HM | No | C+ / B− | S. anginosus group / C. gracilis | Fusobacterium (75.7); S. anginosus group (24.1); Campylobacter (0.17) | 844,388 | A/A |
| 2 | M/53 | L | Yes | Negative | Aspergillus fumigatus | NS[f] | 764 | A/D |
| 3 | M/50 | UK | Yes | C | S. anginosus group | S. anginosus group (72.3); Fusobacterium (25.2); Parvimonas (2.3) | 60,295 | A/A |
| 4 | F/74 | RA | No | C+ | H. influenzae | Haemophilus (99.7) | 108,926 | A/A |
| 5 | M/26 | A | Yes | ND | Negative | NS | 14,275 | A/A |
| 6 | F/82 | Si | No | ND | Aspergillus fumigatus | NS | 1,091 | A/A |
| 7 | M/18 | BSF | No | C+ | S. anginosus group | S. anginosus group (99.9) | 151,310 | A/A |
| 8 | M/72 | L | Yes | Negative | Negative | NS | 6,840 | A/A |
| 9 | M/49 | UK | No | C+ B− | S. anginosus group | S. anginosus group (61.2); Fusobacterium (37.6); Parvimonas (1.2) | 206,029 | A/A |
| 10 | M/67 | Ca | No | C+ | S. anginosus group | S. anginosus group (99.6) | 126,733 | A/D |
| 11 | M/73 | UK | Yes | C+ | S. anginosus group | S. anginosus group (99.8) | 108,784 | A/A |
| 12 | M/70 | UK | Yes | Negative | S. anginosus group | S. anginosus group (57.9); Parvimonas (20.7); Fusobacterium (16.3); Prevotella (3.5); Campylobacter (1.3); Actinomyces (0.3) | 53,180 | A/A |
| 13 | M/50 | HM | Yes | Negative | S. anginosus group | S. anginosus group (99.9) | 78,806 | A/A |
| 14 | F/54 | Hyd | Yes | C+ | S. anginosus group | S. anginosus group (99.8) | 167,461 | A/A |
| 15 | M/11 | UK | No | C+ | S. pneumoniae | S. mitis group (99.7); Fusobacterium (0.13) | 133,666 | A/A |
| 16 | F/63 | UK | No | Negative | A. aphrophilus | Aggregatibacter (99.7); Fusobacterium (0.09); S. anginosus group (0.06) | 99,756 | A/A |
| 17 | M/58 | UK | Yes | B+ | L. monocytogenes | Listeria (99.7); S. anginosus group (0.22) | 57,691 | A/A |
| 18 | F/39 | A | No | C+ | S. anginosus group / A. aphrophilus | S. anginosus group (61.8); Fusobacterium (20.8); Aggregatibacter (15,5); Actinomyces sp. (1.6) | 438,405 | A/A |
| 19 | M/48 | UK | No | B− / B+ / C+ | Fusobacterium sp. / Actinomyces sp. / Prevotella sp. | Fusobacterium (81.9); Porphyromonas (11.2); Catonella (2.6); Treponema (2.3); Peptococcus (0.6); Parvimonas (0.6); S. anginosus group (0.4) | 139,604 | A/A |
| 20 | M/63 | UK | No | Negative | Nocardia sp. | Nocardia (94.4); Ralstonia (2.6) | 18,924 | A/A |
| 21 | M/63 | HC | No | cb+ | L. monocytogenes | Listeria (99.2) | 71,476 | A/A |
| 22 | M/69 | UK | No | C+ | S. anginosus group; A. aphrophilus | Fusobacterium (85.5); S. anginosus group (10.9); Porphyromonas (1.2); Parvimonas (1.23); Campylobacter (0.69); Aggregatibacter (0.18) | 128,429 | A/A |
| 23 | F/70 | Ca | No | C+ | S. anginosus group | S. anginosus group (99.9) | 113,208 | A/D |
| 24 | M/75 | UK | No | C+ B− | S. anginosus group | Fusobacterium (43.8); Prevotella. (26.6); S. anginosus group (26.2); Dialister (3.4) | 128,300 | A/D |
| 25 | F/57 | A | No | C+ | S. anginosus group | S. anginosus group (99.8) | 195,578 | A/A |
| 26 | M/43 | UK | No | Negative | A. actinomycetemcomitans | Fusobacterium (56.2); Campylobacter (36.4); Actinomyces (7.2) | 213,352 | A/A |
| 27 | F/66 | UK | Yes | Negative | Fusobacterium sp. | Fusobacterium (99.9) | 105,644 | A/A |
| 28 | M/69 | UK | No | Negative | Nocardia sp. | Nocardia (68.5); Ralstonia (17.5); Bradyrhizobium (7.2) | 23,658 | A/A |
| 29 | F/71 | UK | No | C+ | A. aphrophilus; Fusobacterium sp.; Parvimonas sp. | Fusobacterium (68.8); Aggregatibacter (18.0); Parvimonas (12.9) | 119,000 | A/A |
| 30 | M/65 | UK | Yes | C+ | S. anginosus group | S. anginosus group (47.5); Fusobacterium (38.8); Parvimonas (11.5); Actinomyces (1.3) | 280,945 | A/A |
| 31 | M/59 | HC | No | C+ / B− B+ | S. anginosus group / Fusobacterium sp. | Fusobacterium (89.0); Parvimonas (10.8); Prevotella. (0.14); S. anginosus group (0.01) | 258,253 | A/A |

**TABLE 2** (Continued)

| Patient no. | Sex/age | Risk factor[a] | Pre-OP AB | Gram stain[c] | Culture | 16S sequencing (% of aligned reads) (genus or group level) | Aligned reads[d] | Outcome at 30 day/ 1 yr[e] |
|---|---|---|---|---|---|---|---|---|
| 32 | F/51 | UK | No | C+ <br> B− | S. anginosus group <br> Campylobacter sp. | Porphyromonas (50.2); Fusobacterium (27.8); S. anginosus group (8.6); Peptostreptococcus (4.4); Tannerella (3.7); Campylobacter (0.93) | 140,897 | A/A |
| 33 | M/34 | UK | No | C+ | A. aphrophilus | Parvimonas (37.4); Aggregatibacter (30.9); Fusobacterium (30.2); Actinomyces (1.0) | 118,415 | A/A |
| 34 | M/43 | Bac | Yes | C+ | S. aureus | S. aureus (99.5); Fusobacterium (0.29) | 162,762 | A/A |
| 35 | M/76 | Bac | Yes | B− | K. pneumoniae | Klebsiella (99.7); Streptococcus (0.20) | 93,519 | A/A |
| 36 | F/53 | UK | Yes | B− <br> C+ | Prevotella sp. <br> Actinomyces sp. | Fusobacterium (54.9); Parvimonas (20.8); Prevotella (13.3); Porphyromonas (3.7); Campylobacter (1.5); Treponema (1.2); Mycoplasma (1.0); Actinomyces sp. (0.7) | 156,010 | A/A |
| 37 | F/74 | UK | No | Negative | Fusobacterium sp. | Fusobacterium (99.9) | 145,204 | A/A |
| 38 | M/0 | PB | Yes | B− | E. coli | Escherichia (99.5); Streptococcus (0.30) | 106,631 | A/A |
| 39 | F/30 | Bac[b] | Yes | Negative | Negative | NS | 17,287 | D/D |
| 40 | F/59 | SS | No | Negative | Negative | NS (identified as the apicoplast genome of T. gondii) | 36,786 | A/D |
| 41 | M/77 | UK | Yes | B+ | L. monocytogenes | Listeria (99.9) | 132,210 | A/A |

[a]A, ethanol and/or cannabinoid abuse; Bac, bacteremia with same microorganism <3 months before; BSF, recent basilar skull fracture; Ca, cancer; Hyd, hydrocephalus after previous brain abscess; HC, hepatic cirrhosis; HM, heart malformation; L, leukemia/myeloma; PB, premature birth and E. coli meningitis; RA, rheumatoid arthritis, methotrexate; Si, chronic sinusitis; SS, systemic sclerosis, immune suppression; UK, none detected.

[b]Bacteremia with S. aureus <3 months before. Surgical concern whether abscess was correctly aspirated.

[c]B−, Gram-negative rods; B+, Gram-positive rods; C, cocci; C+ Gram-positive cocci; cb, coccobacilli; ND, not recorded.

[d]Number of reads with >97% similarity to reference sequences in the SILVA database. Underlined numbers stem from 35 PCR amplification cycles.

[e]A, alive; D, dead.

[f]NS, composition of aligned reads not different from negative controls diagnosed with glioblastoma.

were cultured from two. From 35 samples, 45 bacteria were cultured. A single bacterium was identified in 27 specimens, two in 6, and three bacterial species in 2 specimens; thus, by conventional culture, polymicrobial cultures comprised 8 (22%) of 35 culture-positive samples or 20% of all cases. Monomicrobial culture of *Streptococcus anginosus* group bacteria was challenged by microscopy in two cases, where Gram-negative rods were also observed (patient number 9 and patient number 24) (Table 2).

The cultured bacteria were divided between members of the dental and oropharyngeal microbiota, and bacteria residing elsewhere. Nonoral bacteria were cultured in specimens from eight patients, and these bacteria were only detected in monoculture. *S. aureus* and *Klebsiella pneumoniae* were secondary to bacteremia with the same organisms, while *Escherichia coli* was linked to a case of infant meningitis. More conspicuous were three cases with *Listeria monocytogenes* and two cases with *Nocardia*. Previous bacteremia with these species was not recorded, and a putative introduction to the brain during transient bacteremia indicates a tropism for the cerebral parenchyma. Bacteria originating in the oral cavity were cultured from 27 patients. Bacteria of the *S. anginosus* group were detected in 17 samples; *A. aphrophilus* and the *F. nucleatum* group were identified in five samples each; and *Actinomyces*, *Campylobacter*, and *Prevotella* in two samples each (Table 2).

**Microbial diversity by 16S rRNA gene.** Amplification of the V3-V4 region generated products of the expected size for 31 cases of spontaneous brain abscess. Samples with inadequate product were submitted to 35 PCR cycles, and products from all patients were finally subjected to amplicon sequencing using Illumina metagenomics 16S rRNA technology. Silva database annealing of 175,497 PCR amplification products from nine patients suffering from brain tumors ("negative controls") is described in the methods section. Although most products of negative control samples could not be linked to microbial DNA and probably represent the host, significant proportions aligned with rRNA from *Ralstonia*, *Bradyrhizobium*, *Sphingomonas*, and *Delftia*. PCR products of these species are considered to arise from reagents and handling, and PCR amplifications dominated by these species are regarded as nonsignificant (NS) in assessment of the infective microbiota (Table 2). Four culture-negative specimens and two samples with growth of *Aspergillus* were categorized as NS by amplicon sequencing of 16S rRNA; a special case was patient number 40, where more than 1 million reads belonged to a single sequence that was identified by standard nucleotide BLAST as part of the apicomplexan plastid of *Toxoplasma gondii* (14). Cerebral toxoplasmosis was suspected by histology and confirmed by specific PCR at the Danish reference laboratory.

Combining 5,586,421 aligned reads from 41 samples and using a cutoff value of 0.1% (5,587 reads), PCR amplification of 16S rRNA genes identified the 13 genera recognized by culture plus 7 additional genera (Fig. 1A). But, the relative abundance of reads versus the proportion of isolations were not identical. *Aggregatibacter* and *Actinomyces* were more prominent by culture, while *Fusobacterium* was cultured from 5 of 41 samples but comprised 38% of all reads. *Porphyromonas* was not detected by conventional culture but comprised 1.7% of all reads; *Peptostreptococcus* and *Tannerella* may also be plausible pathogens that escape identification by conventional culture (Fig. 1A). Conversely, a cutoff value of 0.1% of aligned reads (combined abundance of 41 specimens) does not preclude detection of putative bacterial DNA from reagents, such as *Ralstonia*, *Bacillus*, and *Bradyrhizobium* (Fig. 1A, right column).

We also assessed the cumulated number of bacterial detections in 41 samples by applying different cutoffs (Fig. 1B). A breakpoint below 1% was needed to detect *Campylobacter* (cases 1 and 32) (Table 2) and *Actinomyces* (cases 19 and 36) in culture-positive specimens, but low cutoff values generated multiple identifications of uncertain significance. A breakpoint of 0.2% reads per sample identified 57 genera detected in 1 to 20 samples each; restricting calculations to 35 bacterial culture-positive samples reduced the number of genera from 57 to 32, and the cumulated number of detections from 184 to 108 (Fig. 1B).

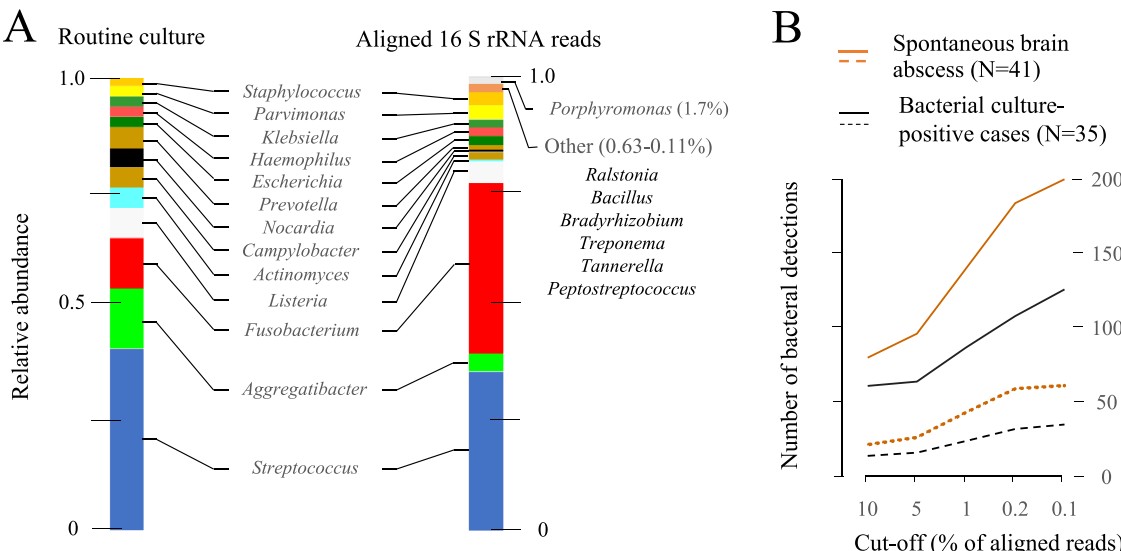

**FIG 1** Bacterial presence in 41 cases of spontaneous brain abscess (identification to genus level). (A) Combined abundance. (Left column) 45 isolations of bacteria from 13 genera (6 samples were bacterial culture negative). (Right column) 20 bacterial genera represented by >0.1% of 5,586,421 aligned reads. (B) Accumulated number of bacterial detections by 16S rRNA gene amplification in 41 samples using different cutoff values. Between 764 and 844,388 reads per sample were aligned with the Silva database; taxa with ≤25 reads were disregarded (this additional breakpoint affected seven samples with 23,658 or fewer reads [see Table 2]). Solid lines, number of bacterial detections; dotted lines, number of different genera. Red color, 41 samples; black color, 35 bacterial culture-positive samples.

We chose a rather conservative cutoff value of 5% of aligned reads per sample to delineate significant presence of bacterial species groups by 16S rRNA gene amplification. This breakpoint is overcautious, but seven monoculture specimens were transformed to polymicrobial infections (cases 3, 9, 12, 24, 26, 30, and 33). The microbiota of six other specimens, which were polymicrobial by culture, were further extended by PCR (cases 1, 18, 29, 31, 32, and 36). Using a cutoff value of 5%, the most common DNA trace of bacteria not detected by culture was *Fusobacterium* and *Parvimonas* (12 and 5 samples, respectively). Table 3 ranks the species groups of spontaneous brain abscess by prevalence, as revealed by conventional culture and cautious interpretation of 16S rRNA gene amplification. Prevalence of the *F. nucleatum* group now equals that of the *S. anginosus* group, and the *Parvimonas micra* group is of the same prominence as genus *Aggregatibacter*.

**Inhibition of culture by antimicrobial treatment.** Patients with intracranial masses were rapidly transferred to the neurosurgical department for treatment, and intravenously infused antimicrobials were initiated before aspiration in less than half of cases (17 of 41). Apparently, the short administration of antimicrobials (less than 24 h) did not significantly impede conventional culture (Table 2); the culture-negative cases (three receiving antimicrobials) were also negative by 16S rRNA gene analysis. Seven patients with spontaneous brain abscess were subjected to repeat aspirations due to relapse or regrowth of abscesses 6 to 33 days after the initial aspiration, four patients with polymicrobial and three with monomicrobial infections (Fig. 2). Conventional culture of repeat aspirations was consistently negative, while amplification of 16S rRNA genes was largely unaffected, although the proportions of species in polymicrobial infections could be affected. The appearance of *Veillonella* and disappearance of the *S. anginosus* group in the second of three aspirations from case number 3 are unexplained. Initial scanning of this patient revealed a solitary occipital abscess, while amplicon sequencing of repeat aspirations could indicate compartmentalization of the mass (Fig. 2). In our case series, 9 patients had multiple abscesses while two solitary abscesses were categorized as multilobular (not shown).

**Reculture of stored abscess material.** In contrast to conventional culture, 16S rRNA analysis promoted the *F. nucleatum* group to the same level of importance as the *S. anginosus*

**TABLE 3** Microbiota of 41 cases of spontaneous brain abscess as revealed by culture, supplemented by amplicon sequencing of 16S rRNA gene fragments from stored specimens[a]

| Species group | No. of positive cases | | | Mono/poly |
|---|---|---|---|---|
| | By culture | By 16S | Total | |
| *Streptococcus anginosus* group | 17 | 16 | 17 | Both |
| *Fusobacterium nucleatum* group | 5[b] | 17 | 17 | Both |
| *Aggregatibacter* species | 6 | 4 | 6 | Both |
| *Parvimonas micra* group | 1[c] | 6 | 6 | Poly |
| *Listeria monocytogenes* | 3 | 3 | 3 | Mono |
| *Prevotella* species | 2[d] | 2 | 3 | Poly |
| *Actinomyces* species | 2[e] | 1 | 3 | Poly |
| *Nocardia* species | 2 | 2 | 2 | Mono |
| *Campylobacter* species | 2 | 1 | 2 | Poly |
| *Aspergillus fumigatus* group | 2 | 0 | 2 | Mono |
| *Porphyromonas* species | 0[f] | 2 | 2 | Poly |
| *Haemophilus influenzae* | 1 | 1 | 1 | Mono |
| *Streptococcus pneumoniae* | 1 | 1 | 1 | Mono |
| *Staphylococcus aureus* | 1 | 1 | 1 | Mono |
| *Escherichia coli* | 1 | 1 | 1 | Mono |
| *Klebsiella pneumoniae* | 1 | 1 | 1 | Mono |
| *Anaerobacillus* species | 0 | 2 | 2 | NR[h] |
| *Cutibacterium* species | 0 | 1 | 1 | NR |
| *Flavobacterium* species | 0 | 1 | 1 | NR |
| *Toxoplasma gondii* | 0 | 1[g] | 1 | Mono |
| Sum | 47 | 64 | 73 | |

[a]Microorganisms are compiled in genera or species groups with the exception of *L. monocytogenes*, five other bacterial species and *Toxoplasma gondii*, which were solely detected in monomicrobial infections and identified by standard methods or monoclonal antibodies, respectively. A cutoff value of 5% of aligned reads/sample was used for species group presence by amplicon sequencing, ignoring *Bacillus*, *Bradyrhizobium*, *Burkholderia*, *Delftia*, *Micrococcus*, *Paracoccus*, *Ralstonia*, *Rhizobium*, and *Sphingomonas*.
[b]Eight additional samples were positive after reculture on selective media.
[c]One additional sample (patient 36) was positive after meticulous reculture of stored specimens.
[d]One additional sample (patient 24) was positive after meticulous reculture of stored specimens.
[e]One additional sample (patient 26) was positive after meticulous reculture of stored specimens.
[f]One sample (patient 19) was positive after meticulous reculture of stored specimens.
[g]16S primers amplified a fragment that could be identified as small subunit rRNA located on the 35-kb apicoplast genome of *Toxoplasma gondii*.
[h]NR, not relevant (only detected in NS samples dominated by negative control species groups [case numbers 2, 6, 8, and 40]).

group. A selective medium that uses vancomycin (for suppression of Gram positives) plus a quinolone (for suppression of many Gram negatives) is commercially available and widely used in Denmark for culture of *Fusobacterium necrophorum* in parapharyngeal aspirations (15). Surplus abscess material was available from 10 patients with spontaneous brain abscess, where microbiota analysis showed significant presence of *Fusobacterium* DNA despite negative culture (including the sample from patient number 9 where initial microscopy identified fusiform rods). After 3 to 8 years of storage at minus 80°C, samples from eight patients (numbers 1, 9, 22, 24, 26, 32, 33, and 36) gave rise to *F. nucleatum* group when plated on *Fusobacterium* selective agar, while samples from 2 patients (numbers 12 and 30) remained negative. Another putative, causative component of brain abscess is the *P. micra* group, which was only detected by routine culture in the sample from patient number 29. *Parvimonas* DNA was present in large amounts in samples from 5 additional patients (11 to 37% of aligned reads) (Table 2), and surplus material was available from 3. Only from one of these was reculture successful, perhaps due to absence of a suitable, selective medium. Meticulous reculture in thioglycolate and on anaerobic agar plates identified additional isolates of *Prevotella*, *Actinomyces*, and *Porphyromonas* (one representative each) (Table 3).

**Categorization and genesis of bacterial spontaneous brain abscess.** Amplicon-based sequencing of 16S rRNA genes supplemented conventional culture, and 35 bacterial infections were finally separated into 15 polymicrobial and 20 monomicrobial

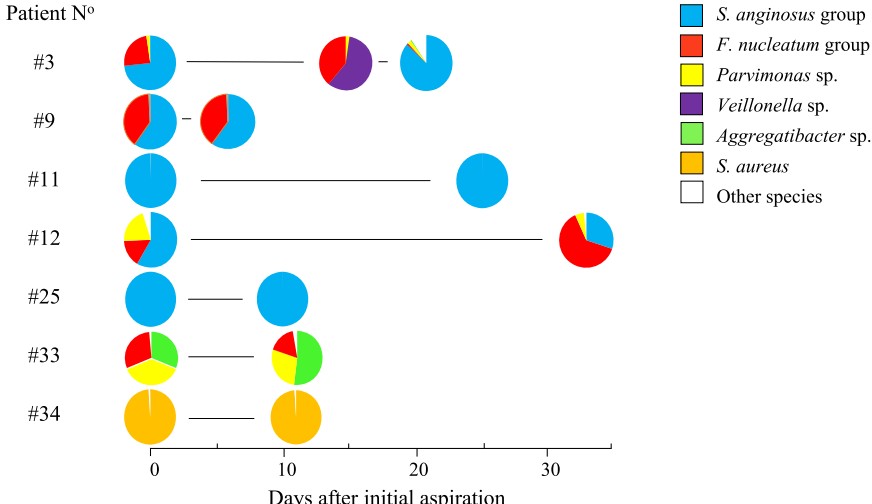

**FIG 2** Distribution of aligned 16S rRNA gene reads in initial and repeated aspirations. Circles represent 15 samples (27,063 to 644,094 reads) from 7 patients. Repeat aspirations were performed between 6 and 33 days after the initial procedure. All repeat aspirations were culture negative.

infections. The former was exclusively caused by resident bacteria of the oral cavity, while monomicrobial infections could be caused by microbiota of adjacent as well as distant habitats. It is rational to presume that polymicrobial infections arise by penetration of the meninges from contiguous foci, while hematogenous dissemination of a single bacterium has the potential to be seeded anywhere in the parenchyma; this theorem is supported by the polybacterial infections of case number 19 (pus from the ear and cholesteatoma), number 24 (close proximity to frontal sinus and ethmoid bone), and number 29 (multiple abscesses near the middle ear). However, anatomical location is not a dependable criterion for identification of polymicrobial infections. Several examples from the present case series show monomicrobial infections in proximity to frontal sinuses (Fig. 3A and B) and polymicrobial infections without contiguity to meninges or sinuses (Fig. 3C and D).

## DISCUSSION

We examined patient records during a 76-month period and identified 41 cases of aspirated spontaneous brain abscess, corresponding to an incidence rate of 0.51 per 100,000 person-years. This rate is less than the estimated nationwide rate of brain abscess of 0.78 (3), but the present study is restricted to patients subjected to surgical drainage and excludes cases with recent neurosurgery. Despite examination of aspirated material by culture, microscopic pathology, and amplicon sequencing, 3 of 41 cases (7%) were unexplained; three other cases were caused by *Aspergillus* or

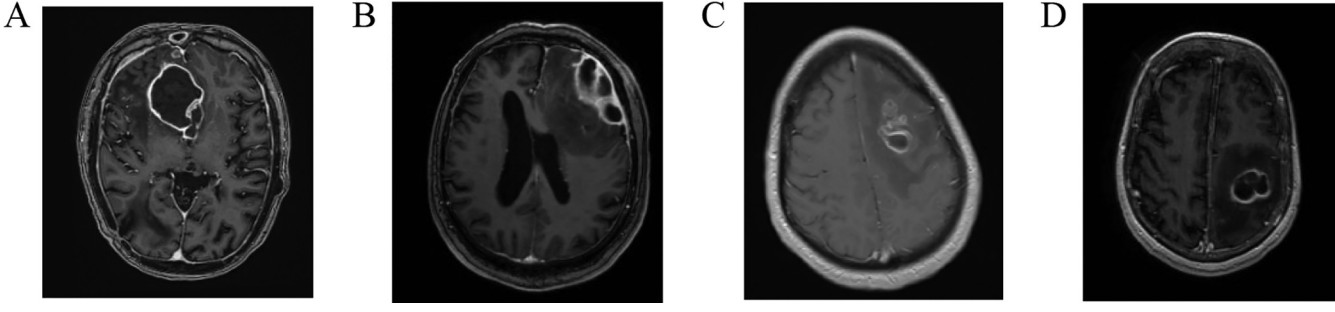

**FIG 3** Selected cases of spontaneous brain abscess caused by the oral microbiota. Patient number 14 (A) and number 11 (B), monomicrobial *S. anginosus* group infections in proximity to frontal sinuses; Patient number 26 (C) and number 32 (D), parenchymal infections with compelling evidence of three or more bacteria but without contiguity to meninges or sinuses. Magnetic resonance image with contrast.

*Toxoplasma*, while 35 cases (85%) were bacterial infections. For at least one of the negative samples, uncertainty of deficient or mistaken puncture was recorded (Table 2).

The causative agent of monobacterial brain abscess is relatively easy to identify by conventional culture if patients have not received prolonged courses of antimicrobial treatment. Our case series is unusual with three cases of *Listeria monocytogenes*, a well-known cause of meningitis and meningoencephalitis that is rarely involved in brain abscess (16). *Nocardia* species are recognized pathogens of brain abscess (1, 12, 17, 18), while the low prevalence of *S. aureus* in the present study probably springs from exclusion of cases with recent brain surgery. Otherwise, our results from conventional culture are typical, with the *S. anginosus* group being the major pathogen, frequent identification of *A. aprophilus* and the *F. nucleatum* group, and significant detection of *Prevotella*, *Campylobacter*, and *Actinomyces*. The molecular pathogenicity of *S. anginosus* group and the conspicuous association with brain infections have been reviewed (19–22).

Spontaneous brain abscess can be caused by bacteria of the oral microbiota or by bacteria residing elsewhere that make use of the bloodstream as a mode of transmission. Embolic events or unrecorded trauma may be involved, and infections originating from distant foci are monomicrobial. In contrast, brain abscess caused by the adjacent microbiota may be monomicrobial or polymicrobial. The challenge for laboratory medicine pertains to polybacterial infections, which may escape detection by conventional culture. Amplicon-based sequencing of 16S rRNA is prone to display an enlarged spectrum of microorganisms in spontaneous brain abscess, but only a limited number of case series have been published (Table 1) and they differ with respect to inclusion criteria, molecular technique, and amplified region of 16S rRNA. Some observations are consistent, such as the expansion of the number of polymicrobial cases, the unexpected importance of *Parvimonas* and *F. nucleatum*, and the repeated detection of *Porphyromonas* (which is exceptionally rare by conventional culture [23]). Other findings are unique and of uncertain significance. *Eubacterium* was detected in 13 of 31 cases using massive parallel sequencing (12), but the finding has not been corroborated. Coagulase-negative staphylococci were considered significant in some studies (11, 13), but these species are notoriously difficult to delineate from procedural contamination during surgery (24). Stebner and coworkers (13) reported *Delftia* in several samples, a genus that is excluded in the present study due to its prominence in negative control specimens.

These differences pinpoint the challenge of establishing a cutoff value for confident detection of pathogens by amplicon-based sequencing. A value below 1% of aligned reads was necessary to identify DNA of some culture-proven species, but a low breakpoint disclosed DNA traces of doubtful significance. Another challenge is allocation of 16S rRNA fragments to species level. A reported distribution (13) within genus *Fusobacterium* by V3-V4 sequencing is questionable, as 14 of 45 *Fusobacterium* detections were attributed to species primarily associated with animals (*Fusobacterium canifelinum*, *Fusobacterium simiae*, and *Fusobacterium equinum*). Further refinements are needed to attain reliable species-level assignment in 16S rRNA gene data sets (25), but bioinformatics refinements and new 16S primer combinations will still be sensitive to possible presence of DNA traces in reagents that can be coamplified. Shotgun metagenomics can accurately identify microorganisms and has the potential to disclose unviable fragments of bacterial DNA, but the biased distribution of host and microbial DNA in abscess material may hamper the general use of such techniques.

Spontaneous brain abscess is a rare but grave infection and a diagnostic challenge to infectious disease specialists and clinical microbiologists. If antimicrobial therapy has been administered for less than 24 h, results from conventional culture and susceptibility testing are dependable, and possible breaches in the prescribed antimicrobial coverage—such as nonsusceptibility of *Listeria* to cephalosporins—can be remedied. The major problem is the identification of the frequent polymicrobial infections. The inadequacy of conventional culture to disclose polymicrobial infections is clearly

indicated by historical records of Gram stain and microscopy of brain abscess material (1, 8); it has been corroborated by amplicon-based sequencing, and it is now verified by culture on selective media. Exclusion of polymicrobial infection by amplicon sequencing can make results from conventional culture irrefutable, and this documentation may be advisable to obtain before discontinuation of, e.g., metronidazole infusion, if only a single agent of oral origin was disclosed by culture. If polymicrobial brain abscess is documented, de-escalation of the initially prescribed broad-spectrum antimicrobial regimen is unlikely, as exhaustive identification of the microbiota by culture is complicated and potentially incomplete.

However, the responsibility of the clinical microbiology laboratory extends beyond the characterization of the single patient case. Certain predisposing and precipitating factors have been described, but the natural history of spontaneous brain abscess must include a description of the causative microbiota. The pathogenesis of the disease is not restricted to incidental access to brain tissue, but involves tropism, cohabitation, aggregation, and persistence in a distinct environment. Selective media advanced the *F. nucleatum* group to the same level of importance as the *S. anginosus* group. When a bacterium is cultured, it can be formally named, fully characterized, manipulated, and subjected to biologic experiments. The peculiar preponderance of certain species or species groups in these infections must be clarified, before a thorough understanding of spontaneous brain abscess can be achieved.

In conclusion, bacteria cause at least 85% of cases of spontaneous brain abscess, and amplicon-based sequencing of 16S rRNA plus culture on selective media can supplement results of conventional culture. Bacterial infections originating from distant foci accounts for one-fifth of all cases and are monomicrobial. Two-thirds of cases are caused by the dental and oropharyngeal microbiota, and most of these infections are polymicrobial. Reliable exclusion of polymicrobial infection is not possible by conventional culture or by anatomical location, and initial antimicrobial therapy must address a broad and mixed spectrum of microorganisms. Culture on selective plates highlights the prominence of *F. nucleatum* group in spontaneous brain abscess, and attention to these bacteria is warranted. *Parvimonas* constitutes another partly neglected genus that must be addressed by the clinical microbiology laboratory.

## MATERIALS AND METHODS

**Patients and routine processing of samples.** The Department of Neurosurgery at Aarhus University Hospital provides brain surgery to inhabitants of the Central Denmark Region with a population of 1,261,000 to 1,304,000 during the study period. Patients with intracranial processes, as revealed by computerized tomography or magnetic resonance imaging, are referred to the department from regional hospitals. Solitary or prominent abscesses with edema and mass effect are drained by aspiration, and samples are distributed to the departments of pathology and clinical microbiology; multiple small masses with minor volume impact are only treated with antimicrobials.

Aspirated material is subjected to Gram staining; plated on 5% horse or sheep blood agar, chocolate agar, anaerobic agar, and selective anaerobic agar (containing 0.01 mg/L kanamycin) plates; and inoculated into serum broth and semisolid thioglycolate vials (SSID, Hillerød, Denmark). Media and plates are incubated at 35°C, the first two plates in the presence of 5% $CO_2$, the latter two plates in an anaerobic atmosphere, and the vials at ambient atmosphere. Negative cultures are dismissed after 7 days of incubation. Cultured microorganisms are identified by phenotypic appearance and matrix-assisted laser desorption ionization–time of flight mass spectrometry (Bruker Daltonics, Bremen, DE); ambiguous identifications are further characterized by Sanger sequencing of the V1-V3 regions of 16S rRNA, as described (26).

From September 2011, surplus abscess material was stored at −80°C for subsequent microbiota analysis.

**Extraction of nucleic acid, PCR, and amplicon sequencing.** Abscess material was thawed and treated with a Precellys lysing kit (Bertin Instruments), and nucleic acids were purified using Magna Pure Compact (Roche Life Science) after treatment with proteinase K, according to manufacturer's instructions. Using the Illumina 16S metagenomic sequencing protocol with primers S-D-Bact-0341-b-S-17 and S-D-Bact-0785-a-A-21 (27, 28), amplicons of approximately 460 bp of the variable V3-V4 region of 16S rRNA were generated with the KAPA HiFi HotStart ReadyMix PCR kit (Kapa Biosystems, South Africa). Standard amplification was denaturation for 5 min at 95°C, followed by 25 cycles of 95°C, 55°C, and 72°C (30 s each) and final elongation at 72°C for 10 min. Negative samples (visually inspected after ethidium bromide staining of gels) were submitted to 35 cycles. PCR-products were purified using magnetic beads (MagSi-NGS[PREP], Amsbio), and 2 × 300-bp reads were generated with MiSeq reagent V3 and Nextera XT library preparation kits (Illumina).

**Bioinformatics analysis.** Bioinformatics analysis was performed on the CLC Genomics Workbench 20.0.2. Adaptor sequences were removed, and trimmed reads were aligned using 97% similarity threshold against the SILVA 16S v132 database (downloaded 18 December 2019, customized by removal of uncultured, ambiguous, unidentified, and mitochondrial sequences). After removal of chimeric sequences, output to genus and species levels were generated for individual samples, and combined relative abundance was calculated for combinations of samples.

**Taxonomic comment/species level identification.** Differentiation between *Streptococcus anginosus*, *Streptococcus constellatus*, and *Streptococcus intermedius* is difficult by standard phenotypic tests and 16S rRNA sequence, and these bacteria are referred to as the *S. anginosus* group. Whole-genome sequencing shows that four subspecies of *Fusobacterium nucleatum* deserve species rank (29), and *F. nucleatum*-related bacteria are referred to as the *F. nucleatum* group. The V3-V4 region of 16S rRNA, in combination with a modest similarity threshold (97%) to the SILVA database, has additional deadlocks, such as inability to identify *Haemophilus influenzae*, *Parvimonas micra*, and *Streptococcus pneumoniae* to the species level. Alignment of reads with SILVA database entries was therefore generally restricted to genus level, unless culture had confirmed species level identification.

**Negative controls for clarification of contaminant DNA.** The reiterative target doublings by PCR can result in amplification of background bacterial DNA from the reagents, even in true-positive samples. To disclose the spectrum of contaminant DNA, we analyzed nine samples with negative culture, where cancer was diagnosed by histologic analysis. The combined relative abundance of 175,497 PCR amplifications (range, 2,945 to 43,527) from nine patients suffering from brain tumors was as follows: no match (71.1%), *Ralstonia* (12.6%), *Bradyrhizobium* (4.3%), *Sphingomonas* (2.8%), *Delftia* (2.2%), *Bacillus* (1.4%), and other genera (5.5%).

**Other methods.** After storage at −80°C for years, selected specimens were thawed and meticulously recultured in thioglycolate semisolid broth and on anaerobic agar plates as described above, plus *Fusobacterium* selective agar plates (SSID, Hillerød, Denmark) containing 5 mg/L nalidixic acid and 2.5 mg/L vancomycin (15).

## ACKNOWLEDGMENTS

We thank Alex L. Laursen, Department of Infectious Medicine, Aarhus University Hospital, for contribution to disease categorization.

The study received internal funding from Aarhus University Hospital, Sparrow head pool 2015. The study was conducted in accordance with national guidelines for the use of clinical and laboratory data and was approved by the judicial office, Central Denmark Region (record number 1-16-02-948-17) and the National Board of Health (record number 3-3013-2398/1).

N.N.-L. conceived and planned the study and wrote the first draft of the manuscript. W.R. improved molecular analysis, and C.A. devised supplementary culture. B.B. categorized infections and evaluated outcome. All authors contributed significant intellectual input to the manuscript and approved the final draft.

We declare that there are no conflicts of interests.

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
