## [Reviewer comments · Microbiology Spectrum]

Microbiology Spectrum

Culture on selective media and amplicon-based sequencing of 16S rRNA from spontaneous brain abscess – the view from the diagnostic laboratory

Camilla Andersen, Bo Bergholt, Winnie Ridderberg, and Niels Nørskov-Lauritsen

Corresponding Author(s): Niels Nørskov-Lauritsen, Odense University Hospital

Review Timeline:

Submission Date:	December 1, 2021
Editorial Decision:	December 29, 2021
Revision Received:	March 9, 2022
Accepted:	March 10, 2022

Editor: Jennifer Dien Bard

Reviewer(s): Disclosure of reviewer identity is with reference to reviewer comments included in decision letter(s). The following individuals involved in review of your submission have agreed to reveal their identity: Henrik Nielsen (Reviewer #1)

Transaction Report:

DOI: <https://doi.org/10.1128/spectrum.02407-21>

December 29, 2021

Prof. Niels Nørskov-Lauritsen
Odense University Hospital
Clinical Microbiology
Odense C
Denmark

Re: Spectrum02407-21 (Culture on selective media and amplicon-based sequencing of 16S rRNA from spontaneous brain abscess - the view from the diagnostic laboratory)

Dear Prof. Niels Nørskov-Lauritsen:

Thank you for submitting your manuscript to Microbiology Spectrum. The manuscript was reviewed by two expert reviewers and they both agree that major revisions are required and must be appropriately addressed before it can be considered for publication. When submitting the revised version of your paper, please provide (1) point-by-point responses to the issues raised by the reviewers as file type "Response to Reviewers," not in your cover letter, and (2) a PDF file that indicates the changes from the original submission (by highlighting or underlining the changes) as file type "Marked Up Manuscript - For Review Only". Please use this link to submit your revised manuscript - we strongly recommend that you submit your paper within the next 60 days or reach out to me. Detailed instructions on submitting your revised paper are below.

Link Not Available

Sincerely,

Jennifer Dien Bard

Journals Department
Reviewer comments:

Reviewer #1 (Comments for the Author):

The microbiological diagnosis of brain abscess material may be challenging and culture is tedious in mixed cultures. Oral cavity flora are often regarded as the most prevalent but the detailed number of isolates and species are not always known. By use of novel technologies, including PCR amplification of bacterial genes, a more complete diagnostic report may potentially be possible, including information relevant for clinical care. The authors examined a large number of archived samples from brain abscess aspirations and subjected the material to expanded investigations with the aim to increase the information of microbiological representation in the samples.

The main conclusion is, by use of metagenomic analyses and of selective media for culture, that brain abscess is of polymicrobial nature rather than monomicrobial. The study is adding important knowledge to the biological understanding of brain abscess. However, some amendments to the manuscript could improve the communication.

1. The authors make an interesting observation in Figure 3, that the anatomical location of the abscess (proximity to sinuses yes/no) and the possible association with monomicrobial vs polymicrobial findings. The authors should report the distribution in this regard for all patients calculating if the anatomical location predicts the probability of monomicrobial (n=26) vs polymicrobial (n=15) findings

2. Notably, with non-oral bacteria detected by culture all cases had monoculture (lines 146-148), whereas oral bacteria had many cases with polymicrobial diagnosis. This fact could be more prominently discussed in DISCUSSION. Could it relate to different pathogenesis ?
3. A reference is made to Al Masselma (line 268) from 2009 regarding Mycoplasma species in brain abscess and another reference (Ørsted et al, Clin Microbiol Infection 2011; 17: 1047-9) could be added to support Mycoplasma as a potential pathogen in brain abscess
4. The main findings of the study (frequent occurrence of polymicrobial abscess) is in agreement with three previous studies as listed in Table 1. The authors should include in the discussion/conclusion how this observation could impact on the choice of clinical care and antimicrobial chemotherapy of brain abscess
5. In Table 1 is listed the number of cases of polymicrobial infections - it is recommended to add the percentages of such polymicrobial infections for each of the studies

Reviewer #2 (Comments for the Author):

This paper presents a retrospective analysis on the culture and sequence of spontaneous brain abscesses in 41 patients. The methodology and execution of the study is well thought out and designed well. This is an important piece of work adding to the knowledge on the microbiota of spontaneous brain abscesses. However, I feel that the discussion of the results and the conclusions made can be more impactful. The authors did a good job of describing the data being seeing but not interpreting it into the big picture. Why are these data important? Are they going to make a difference into the practice of clinical microbiology or impact patient care? There was a statement of important insights into the nature of polymicrobial spontaneous brain abscesses but I do not know if they were fully conveyed in the discussion.

My edits and suggestions are listed below:

Line 35: Define ICD-10

Line 52: Define PCR

Line 190: Table 3 ranks the species identification of...

Line 194: Define IV

Line 231: Figure 3- Are these patients in the actual study? If so, this needs to be stated in the results and within the figure legend. I feel like it would be nice to see the corresponding culture and sequencing profiles of the abscess that is shown for each patient. If they are not patients described in this study, then this figure needs to be removed since they are not discussed in the paper.

DISCUSSION AND CONCLUSION: There needs to be elaboration on what the data is showing and input into the discussion and possible suggestions for clinical microbiology and the impact it would have on patient care is needed.

Can you explain the cases where culture showed growth of an organism, but it was only detected below 1% reads? (Patients 22, 31, 32 and 36) What does this mean for sequencing? Is the cultured organism the most important or are the other organisms sequenced in the sample more important?

Should this sequencing of spontaneous brain abscess be used for clinical care? Why or why not? Would it have a positive or negative impact on patient/clinical care.

Do you recommend adding the Fusobacterium selective plate to the culturing of these abscesses? Do you think this addition of selective media would have a positive or negative impact on patient/clinical care?

Staff Comments:

Preparing Revision Guidelines

- Point-by-point responses to the issues raised by the reviewers in a file named "Response to Reviewers," NOT IN YOUR

COVER LETTER.

- Upload a compare copy of the manuscript (without figures) as a "Marked-Up Manuscript" file.
- Each figure must be uploaded as a separate file, and any multipanel figures must be assembled into one file.
- Manuscript: A .DOC version of the revised manuscript
- Figures: Editable, high-resolution, individual figure files are required at revision, TIFF or EPS files are preferred

Please return the manuscript within 60 days; if you cannot complete the modification within this time period, please contact me. If you do not wish to modify the manuscript and prefer to submit it to another journal, please notify me of your decision immediately so that the manuscript may be formally withdrawn from consideration by Microbiology Spectrum.

Reviewer #1 (Comments for the Author):

The microbiological diagnosis of brain abscess material may be challenging and culture is tedious in mixed cultures. Oral cavity flora are often regarded as the most prevalent but the detailed number of isolates and species are not always known. By use of novel technologies, including PCR amplification of bacterial genes, a more complete diagnostic report may potentially be possible, including information relevant for clinical care. The authors examined a large number of archived samples from brain abscess aspirations and subjected the material to expanded investigations with the aim to increase the information of microbiological representation in the samples.

The main conclusion is, by use of metagenomic analyses and of selective media for culture, that brain abscess of is of polymicrobial nature rather than monomicrobial. The study is adding important knowledge to the biological understanding of brain abscess. However, some amendments to the manuscript could improve the communication.

1. The authors make an interesting observation in Figure 3, that the anatomical location of the abscess (proximity to sinuses yes/no) and the possible association with monomicrobial vs polymicrobial findings. The authors should report the distribution in this regard for all patients calculating if the anatomical location predicts the probability of monomicrobial (n=26) vs polymicrobial (n=15) findings

Reply. The question of proximity to meninges and adjacent focal infections is not straightforward in all cases. Figure 3 shows 4 cases in conflict with the rational presumption of a link between anatomical location and microbiota, and the patient cases are now specified. These cases invalidate a general acceptance of the theorem.

2. Notably, with non-oral bacteria detected by culture all cases had monoculture (lines 146-148), whereas oral bacteria had many cases with polymicrobial diagnosis. This fact could be more prominently discussed in DISCUSSION. Could it relate to different pathogenesis ?

Reply. The last part of the Discussion section has been expanded to draw attention to this fact.

3. A reference is made to Al Masselma (line 268) from 2009 regarding Mycoplasma species in brain abscess and another reference (Ørsted et al, Clin Microbiol Infection 2011; 17: 1047-9) could be added to support Mycoplasma as a potential pathogen in brain abscess

Reply. The reviewer highlights an important reference to Mycoplasma sp., but that part of the Discussion section has been removed in order to make the overall conclusions more impactful.

4. The main findings of the study (frequent occurrence of polymicrobial abscess) is in agreement with three previous studies as listed in Table 1. The authors should include in the discussion/conclusion how this observation could impact on the choice of clinical care and antimicrobial chemotherapy of brain abscess

Reply. We have rewritten the second part of the Discussion section, plus the Conclusion, to address the concerns raised by the reviewer.

5. In Table 1 is listed the number of cases of polymicrobial infections - it is recommended to add the percentages of such polymicrobial infections for each of the studies

Reply. Percentages of polymicrobial infections have been inserted in Table 1.

Reviewer #2 (Comments for the Author):

This paper presents a retrospective analysis on the culture and sequence of spontaneous brain abscesses in 41 patients. The methodology and execution of the study is well thought out and designed well. This is an important piece of work adding to the knowledge on the microbiota of spontaneous brain abscesses. However, I feel that the discussion of the results and the conclusions made can be more impactful. The authors did a good job of describing the data being seeing but not interpreting it into the big picture. Why are these data important?

Are they going to make a difference into the practice of clinical microbiology or impact patient care? There was a statement of important insights into the nature of polymicrobial spontaneous brain abscesses but I do not know if they were fully conveyed in the discussion.

My edits and suggestions are listed below:

Line 35: Define ICD-10

Reply. The abbreviation has been written out

Line 52: Define PCR

Reply. The abbreviation has been written out

Line 190: Table 3 ranks the species identification of...

Reply. The Table ranks species groups according to prevalence. This is now specified in the text.

Line 194: Define IV

Reply. The abbreviation has been written out.

Line 231: Figure 3- Are these patients in the actual study? If so, this needs to be stated in the results and within the figure legend. I feel like it would be nice to see the corresponding culture and sequencing profiles of the abscess that is shown for each patient. If they are not patients described in this study, then this figure needs to be removed since they are not discussed in the paper.

Reply. Figure 3 shows 4 cases in conflict with the rational presumption of a link between anatomical location and microbiota. The patient cases are now specified, and culture and sequencing profiles are therefore available in Table 2.

DISCUSSION AND CONCLUSION: There needs to be elaboration on what the data is showing and input into the discussion and possible suggestions for clinical microbiology and the impact it would have on patient care is needed.

Reply. We have rewritten the second part of the Discussion section, plus the Conclusion, to address the concerns raised by the reviewer. Particularly, the challenge to the diagnostic laboratory extends beyond characterization of the individual patient case.

Can you explain the cases where culture showed growth of an organism, but it was only detected below 1% reads? (Patients 22, 31, 32 and 36) What does this mean for sequencing? Is the cultured organism the most important or are the other organisms sequenced in the sample more important?

Reply. Amplicon-based sequencing is a powerful supplement to culture, but the method is not without pitfalls. We have briefly discussed ambiguities and controversies between reported results, and we make short mention of shotgun sequencing. In the last paragraph of the Discussion section, we emphasize that amplicon-based sequencing cannot replace culture in order to obtain a thorough understanding of the infection.

Should this sequencing of spontaneous brain abscess be used for clinical care? Why or why not? Would it have a positive or negative impact on patient/clinical care.

Reply. We have rewritten the last part of the Discussion section to highlight the importance of amplicon-based sequencing. One critical contribution will be to exclude polymicrobial infections, in which case the results from conventional culture and antimicrobial susceptibility testing are irrefutable.

Do you recommend adding the Fusobacterium selective plate to the culturing of these abscesses? Do you think this addition of selective media would have a positive or negative impact on patient/clinical care?

Reply. Presence of *Fusobacterium* must be suspected in spontaneous brain abscess, and initial therapy must cover these bacteria. The true value of Fusobacterium selective agar plates is to enable characterization of this group, and thereby facilitate a proper understanding of a rare, but grave infection. We have tried to emphasize this aspect in the final part of the Discussion section.

March 10, 2022

Prof. Niels Nørskov-Lauritsen
Odense University Hospital
Clinical Microbiology
Odense C
Denmark

Re: Spectrum02407-21R1 (Culture on selective media and amplicon-based sequencing of 16S rRNA from spontaneous brain abscess - the view from the diagnostic laboratory)

Dear Prof. Niels Nørskov-Lauritsen:

I am pleased to share that your manuscript has been accepted, and I am forwarding it to the ASM Journals Department for publication. You will be notified when your proofs are ready to be viewed.

Sincerely,

Jennifer Dien Bard
Editor, Microbiology Spectrum
